# The Dipeptide Gly-Pro (GP), Derived from *Hibiscus sabdariffa*, Exhibits Potent Antifibrotic Effects by Regulating the TGF-β1-ATF4-Serine/Glycine Biosynthesis Pathway

**DOI:** 10.3390/ijms241713616

**Published:** 2023-09-03

**Authors:** HaiVin Kim, YoungSu Jang, JaeSang Ryu, DaHye Seo, Sak Lee, SungSoo Choi, DongHyun Kim, SangHyun Moh, JungU Shin

**Affiliations:** 1Department of Biomedical Science, College of Life Science, Graduate School, CHA University, Seongnam 13488, Republic of Korea; kimteam01@naver.com (H.K.); sssjus@naver.com (Y.J.); dahye5448@naver.com (D.S.); 2Department of Dermatology, CHA Bundang Medical Center, CHA University School of Medicine, Seongnam 13496, Republic of Korea; rjsang7496@naver.com (J.R.); terios92@hanmail.net (D.K.); 3Plant Cell Research Institute of BIO-FD&C Co., Ltd., Incheon 21990, Republic of Korea; slee@biofdnc.com; 4Daesang Holdings, Jung-gu, Seoul 04513, Republic of Korea; schoi@daesang.com

**Keywords:** *Hibiscus sabdariffa*, TGF-β1, de novo serine/glycine biosynthesis, fibrosis

## Abstract

TGF-β1, a key fibrotic cytokine, enhances both the expression and translocation of the activating transcriptional factor 4 (ATF4) and activates the serine/glycine biosynthesis pathway, which is crucial for augmenting collagen production. Targeting the TGF-β1-ATF4-serine/glycine biosynthesis pathway might offer a promising therapeutic approach for fibrotic diseases. In this study, we aimed to identify a proline-containing dipeptide in *Hibiscus sabdariffa* plant cells that modulates collagen synthesis. We induced *Hibiscus sabdariffa* plant cells and screened for a proline-containing dipeptide that can suppress TGF-β1-induced collagen synthesis in fibroblasts. Analyses were conducted using LC-MS/MS, RT-qPCR, Western blot analysis, and immunocytochemistry. We identified Gly-Pro (GP) from the extract of *Hibiscus sabdariffa* plant cells as a dipeptide capable of suppressing TGF-β1-induced collagen production. GP inhibited the phosphorylation of Smad2/3 and reduced the expression of ATF4, which is upregulated by TGF-β1. Notably, GP also decreased the expression of enzymes involved in the serine/glycine biosynthesis and glucose metabolism pathways, such as *PHGDH, PSAT1, PSPH, SHMT2,* and *SLC2A1*. Our findings indicate that the peptide GP, derived from *Hibiscus sabdariffa* plant cells, exhibits potent anti-fibrotic effects, potentially through its regulation of the TGF-β1-ATF4-serine/glycine biosynthesis pathway.

## 1. Introduction

TGF-β1, a key cytokine, has a significant role in the development of fibrosis, a pathological condition marked by an excessive accumulation of extracellular matrix proteins in tissues, resulting in organ dysfunction [1]. Beyond its recognized influence on cell growth, differentiation, cellular homeostasis, and extracellular matrix synthesis, TGF-β1 also profoundly affects cellular metabolism [2], encompassing the regulation of glucose, lipid, and amino acid metabolism [3]. One pivotal method by which TGF-β1 orchestrates cellular metabolism involves the regulation of activating transcription factor 4 (ATF4) [4,5], a transcription factor implicated in various cellular stress responses, such as oxidative stress [6], endoplasmic reticulum stress [7], and nutrient deprivation [8]. Moreover, ATF4 is instrumental in collagen synthesis in fibroblasts and myofibroblasts, promoting the expression of genes associated with amino acid metabolism like phosphoglycerate dehydrogenase (PHGDH), phosphoserine phosphatase (PSPH), phosphoserine aminotransferase-1 (PSAT-1), and serine hydroxymethyltransferase (SHMT), thereby providing glycine for augmented collagen production in reaction to fibrotic stimuli [5]. Consequently, targeting the TGF-β1/ATF4-driven activation of the serine/glycine synthesis pathway emerged as a promising avenue for fibrotic disease treatment [4,5].

*Hibiscus sabdariffa*, also known as Roselle, harbors beneficial compounds such as anthocyanins [9,10], flavonoids [11,12], and organic acids [13], displaying anti-cancer [14] and anti-inflammatory attributes [15]. As a result, *Hibiscus sabdariffa* is widely used in a diverse range of products, including processed foods, beverages, jellies, confectionery, and pharmaceutical preparations [16]. However, further research is needed to fully understand the precise mechanisms by which *Hibiscus sabdariffa* executes its biological functions.

Utilizing plant cell culture technology facilitates continuous production of biomaterials year-round while minimizing the environmental impact by harnessing cultured plant cells instead of extracting directly from the plants. Thus, we induced plant cells from *Hibiscus sabdariffa* and explored substances within its cell extract that could modulate collagen synthesis. We identified Gly-Pro (GP), a dipeptide, which exhibited anti-fibrotic activity in fibroblasts. GP inhibited TGF-β1-induced collagen synthesis by suppressing Smad2/3 phosphorylation, down-regulating ATF4, and modulating serine/glycine biosynthesis pathways.

## 2. Results

### 2.1. Identification of Hibiscus Sabdariffa-Derived Dipeptides

Plant cells were mass-cultured from the young leaves of *Hibiscus sabdariffa* and subsequently dried (Figure 1a). It is well established that proline and hydroxyproline are amino acids constituting collagen [17,18,19], and proline-containing dipeptides may influence the biological function of fibroblasts [20]. Accordingly, the extract of *Hibiscus sabdariffa* plant cells was analyzed to detect the presence of dipeptides containing proline or hydroxyproline. This analysis was performed by monitoring the molecular ion (precursor ion) or daughter ion peaks of tandem mass spectra. Based on this monitoring, we identified the following peptides in the extract: Ala-Pro(AP), Gly-Pro(GP), Asn-Pro(NP), Arg-Pro(RP), Ser-Pro(SP), Val-Pro(VP), Tyr-Pro(YP), Asn-Hyp(NO), Thr-Hyp(TO), Val-Hyp(VO), and Trp-Hyp(OW). These 11 peptides were individually synthesized and utilized as analytical standard compounds for both qualitative and quantitative analysis. The MS/MS spectra of each peak observed in the extract of *Hibiscus sabdariffa* plant cells were compared to the MS/MS spectra of each standard compound to determine whether their patterns matched. It was confirmed that AP, NP, RP, SP, VP, YP and GP were present in the extract of *Hibiscus sabdariffa* plant cells, while NO, TO, VO, and OW were not detected (Figure 1b,c, and Appendix A). Moreover, the quantities of each peptide within the *Hibiscus sabdariffa* plant cell extract were measured under multi-reaction monitoring mode (Figure 1d and Appendix A). The respective contents were found to be as follows: AP 13.56 µg/g, GP 6.2 µg/g, NP 5.12 µg/g, RP 21.72 µg/g, SP 10.72 µg/g, VP 14.96 µg/g, and YP 7.08 µg/g.

### 2.2. A Screening for a Dipeptide Suppressing TGF-β1-Induced Collagen Synthesis in Fibroblasts

Next, we aimed to identify peptides capable of mitigating the overexpression of *COL1A1* and *COL3A1* in response to TGF-β1, a key fibrotic cytokine. With a focus on peptides that might modulate collaen synthesis, we treated human dermal fibroblasts (HDF) with the seven previously identified dipeptides along with TGF-β1. We then evaluated whether these peptides could suppress the TGF-β1-induced increase in collagen synthesis. Among the peptides examined, GP emerged as a promising candidate, demonstrating a substantial ability to reduce the expression levels of *COL1A1* and *COL3A1,* as depicted in Figure 2a,b.

### 2.3. GP Suppressed TGF-β1-Induced Smad2/3 Phosphorylation and Collagen Synthesis

TGF-β1, a critical cytokine in fibrotic diseases, activates the Smad2/3 complexes and, in conjunction with the co-mediator Smad4, these complexes translocate to the nucleus. Once inside the nucleus, they bind to Smad binding elements (SBEs) within collagen gene promoters, initiating the transcription of collagen genes [21]. In this context, we explored whether GP could suppress Smad2/3 phosphorylation and whether it could reduce TGF-β1-induced collagen synthesis at both the mRNA and protein level. Following treatment of HDF cells with TGF-β1, robust phosphorylation of Smad2/3 was observed (Figure 3a), and GP markedly inhibited this phosphorylation (Figure 3b). RT-qPCR analysis revealed that GP significantly decreased the mRNA expression of *COL1A1, COL1A2*, and *COL3A1*, which had been elevated by TGF-β1 (Figure 3c–e). To validate these observations at the protein level, we conducted immunocytochemistry (ICC) assays. ICC analyses confirmed that the protein expression of Collagen type 1, elevated by TGF-β1, was subsequently reduced by GP (Figure 3f,g). Moreover, in the supernatant of HDFs treated with both GP and TGF-β1, the expression of pro-collagen 1 was found to be lower compared to that in the supernatant of HDFs treated with TGF-β1 alone (Figure 3h).

### 2.4. GP Downregulated ATF4 Expression

TGF-β1 is also known to promote the expression of ATF4, a protein involved in various cellular processes, including cellular metabolism and growth [5]. Notably, ATF4 plays a vital role in enhancing the expression of enzymes within the serine/glycine synthesis pathway, enabling augmented collagen synthesis [22]. Therefore, we next sought to investigate whether GP affects the expression of ATF4. Both RT-qPCR and Western blot analysis demonstrated that GP inhibited ATF4 expression at the mRNA and protein levels, as shown in Figure 4a,b. Moreover, upon TGF-β1 stimulation, ATF4 undergoes nuclear translocation and then modulates the transcription of its target genes [5,23]. In the ICC analyses, TGF-β1 significantly increased the expression of ATF4 in the nucleus, and GP markedly reduced its expression (Figure 4c).

### 2.5. GP Decreased the Expression of Enzymes Involved in Serine/Glycine Biosynthesis

ATF4 plays a pivotal role in the de novo biosynthesis of serine and glycine, which are essential precursors for collagen production. The initial step in this pathway involves PHGDH converting 3-phosphoglycerate into phosphohydroxypyruvate. Subsequently, PSAT1 forms phosphoserine by transferring an amino group to phosphohydroxypyruvate. Then, PSPH hydrolyzes phosphoserine to produce serine, and finally, SHMT transforms serine into glycine by transferring a one-carbon unit to tetrahydrofolate, resulting in glycine and 5,10-methylenetetrahydrofolate [4,5]. These enzymes work in synergy to ensure sufficient serine and glycine synthesis, which is fundamental to collagen biosynthesis in fibroblasts [5,24]. Moreover, ATF4 boosts the expression of genes such as glucose transporter 1 (GLUT1) that cooperate with the serine/glycine biosynthetic enzymes to enhance glycine synthesis from glucose, further supporting collagen production [5].

In our study, we sought to explore the regulatory role of GP on these key enzymes within the serine/glycine biosynthesis and glucose metabolism pathways, specifically PHGDH, PSAT1, PSPH, SHMT2, and GLUT1. We conducted RT-qPCR analysis, and our findings revealed that TGF-β1 significantly upregulated the gene expression of these enzymes. In alignment with our previous observations, GP was shown to decrease the TGF-β1-induced elevation of *PHGDH, PSAT1, PSPH, SHMT2*, and *SLC2A1* (Figure 5a–e), underscoring the potential of GP for modulating serine and glycine synthesis, and thereby collagen production.

## 3. Discussion

Fibrosis is a pathological process characterized by the excessive accumulation of collagen and other extracellular matrix components within tissues, leading to the disruption of tissue structure and potential organ dysfunction [25,26]. Particularly in the skin, conditions such as hypertrophic scars and keloids may occur due to abnormal fibrosis during the wound healing process [27]. However, the means to treat or prevent these complications remain limited. Recent advancements have underscored the significance of metabolic reprogramming in the activation and function of fibroblasts, the primary cells responsible for producing extracellular matrix components [28,29]. These metabolic adaptations provide the essential energy and building blocks required for collagen synthesis and other fibrotic activities [29]. Consequently, targeting these metabolic pathways could offer a promising therapeutic strategy for the management of fibrotic diseases. In this research, we aimed to identify substances from *Hibiscus sabdariffa*, a plant renowned for its anti-cancer properties [14] and antioxidant activity [11], that are capable of regulating these metabolic pathways. Among the proline-containing dipeptides in the *Hibiscus sabdariffa* plant cell extract, we discovered that GP effectively reduced collagen gene expression augmented by TGF-β1. This modulation occurred through the inhibition of TGF-β1-induced phosphorylation of Smad2/3, thereby influencing the ATF4-mediated activation of the serine/glycine pathway.

ATF4 is a pivotal mediator in maintaining metabolic homeostasis and oxidative equilibrium [6]. Within the context of fibrosis, it plays a critical role in directing the synthesis of glycine, an essential amino acid that is fundamental to collagen formation [30]. The transcription and translation of ATF4 can be modulated by TGF-β1, a key mediator of tissue fibrosis. Smad3, one of the primary downstream regulators of TGF-β1 signaling, interacts with the ATF4 promoter region, thus influencing its transcriptional activity [5,31]. While TGF-β1-activated Smad3 governs ATF4 transcription, TGF-β1 also amplifies ATF4 translation through the mTORC1-4E-BP1 axis [5]. In our study, we demonstrated that TGF-β1 increased Smad2/3 phosphorylation and ATF4 expression, and GP markedly inhibited TGF-β1-induced Smad2/3 phosphorylation and suppressed ATF4 expression. These observations imply that GP possesses the capacity to impede the downstream signaling of TGF-β1, leading to a consequent reduction in ATF4 expression. Such a modulation may further influence collagen synthesis by orchestrating the de novo serine/glycine biosynthesis pathway.

Glycine, a primary component of collagen—the predominant structural protein in connective tissues—directly influences collagen production [32]. Thus, managing glycine synthesis could be instrumental in mitigating excessive collagen deposition [1]. ATF4 is known to govern the biosynthesis of serine, a precursor to glycine. The de novo serine/glycine biosynthesis pathway is a complex series of enzymatic reactions carried out by specialized enzymes, including PHGDH, PSAT1, and PSPH, all under the regulation of ATF4 [5]. This chain of reactions ultimately leads to the conversion of serine to glycine via SHMT [5,24]. Additionally, ATF4 enhances the expression of GLUT1, promoting glycine synthesis from glucose and further supporting collagen production [5]. Consequently, TGF-β1-induced ATF4 modulates the expression of these enzymes, thereby contributing to increased collagen synthesis [4,5]. Our findings, which demonstrate that GP suppresses the TGF-β1-induced increase in these enzymes, highlight the role of GP in regulating the serine/glycine biosynthesis pathway as well as ATF4. By inhibiting the activity of these pivotal enzymes, GP holds its potential to modulate collagen synthesis and attenuate fibrogenesis.

In summary, our study suggests that GP, derived from *Hibiscus sabdariffa* plant cells, suppresses ATF4 expression via the Smad2/3 pathway, resulting in the inhibition of TGF-β1-induced collagen synthesis. Furthermore, GP regulates the de novo serine/glycine pathway, which is essential for excessive collagen synthesis in response to TGF-β1. Considering that amino acids such as serine and glycine are not only foundational to collagen biosynthesis but also to vital cellular functions [30], the implication that GP may control this pathway reveals a potential multifaceted role in cellular metabolism, extending beyond mere collagen modulation and fibrosis.

## 4. Materials and Methods

### 4.1. Hibiscus Sabdariffa Plant Cell Extract

*Hibiscus sabdariffa* plant cells were induced from the leaf tissue. The induced plant cells were mass cultured and dried. *Hibiscus sabdariffa* plant cell extract was prepared via ultrasonic extraction for 2 h by adding 40 times water to dried *Hibiscus sabdariffa* plant cells. The extract was filtered and used for analysis.

### 4.2. LC-MS/MS Analysis

Qualitative and quantitative analysis of low molecular weight dipeptides containing proline or hydroxyproline, an amino acid that constitutes collagen, was performed using an AB SCIEX 3200 QTRAP MS/MS (Applied Biosystems, Waltham, MA, USA) equipped with a 1200 Series HPLC system (Agilent Technologies, Santa Clara, CA, USA). It was monitored through the EPI (enhanced product ion) mode in electrospray ionization positive mode for the qualitative analysis of peptides, and the quantitative analysis was performed in MRM (multi-reaction monitoring) mode. The conditions for instrumental analysis are described in Appendix A. We synthesized the peptides and used them as analytical standard compounds in order to perform qualitative and quantitative analysis of the peptides predicted to be present in the extract. Each peptide was synthesized via solid phase peptide synthesis, and the purity was over 95%.

### 4.3. Antibodies

p-Smad2/3, Smad2/3, ATF4, antibodies were purchased from Cell Signaling Technology (Danvers, MA, USA). Beta-actin antibody was purchased from Sigma-Aldrich, St. Louis, MO, USA. Collagen type 1 antibody was purchased from Novus Biologicals (Littleton, CO, USA). Pro-collagen 1 antibody was purchased from abcam (Cambridge, UK).

### 4.4. Cell Culture and Treatment

Human dermal fibroblasts (HDF) were isolated and cultured from a foreskin. Experiments were performed with using HDF at passages 6 and 7. HDF cells were cultured in Dulbecco’s modified Eagle’s medium (DMEM; Welgene, Daegu, Republic of Korea) supplemented with a 10% fetal bovine serum (FBS, Welgene) and 1% P/S solution (Welgene). HDFs were starved for 24 h prior to stimulation with 1 ng/mL of TGF-β1 (Peprotech, Cranbury, NJ, USA) or 6 µg/mL of GP in serum-free medium.

### 4.5. Western Blot

Cells were harvested, washed in PBS, and lysed in a lysis buffer (Intronbio, Seongnam, Republic of Korea). The protein content of the supernatant was quantified using Quick Start™ Bradford 1 × Dye Reagent (Bio-Rad, Hercules, CA, USA). Samples were separated by SDS-PAGE and transferred onto the polyvinylidene fluoride (PVDF) membranes (Sigma-Aldrich). Blots were detected by adding Clarity Western ECL Substrate (Bio-Rad). For secreted proteins, HDF culture supernatants were collected and concentrated with Vivaspin 6 centrifugal concentrators (Sartorius, Göttingen, Germany). Loaded samples were prepared with 5X sample buffer. Beta-actin was used as a loading control.

### 4.6. Quantitative PCR

Total RNA was isolated from the cells by using Trizol (Invitrogen, Carlsbad, CA, USA). cDNA was synthesized using oligo dT primers and M-MLV reverse transcriptase (Promega Corporation, Madison, WI, USA) and oligo dT primers (Bioneer, Daejeon, Republic of Korea). Relative mRNA expression was quantified via real-time PCR using SYBR Green master mix (Bioneer). In quantitative real-time PCR, GAPDH was used for normalization following the 2^−ΔΔCT^ method.

### 4.7. Immunocytochemistry

HDF cells were seeded on glass-cover dishes and washed with PBS and then fixed in 4% paraformaldehyde. The cells were then treated with 0.1% Triton-X-100 (Sigma-Aldrich) in PBS for 10 min and blocked with 1% BSA in PBST for 45 min at room temperature. The samples were then incubated with primary antibodies at a dilution of 1:100 overnight at 4 °C, followed by secondary antibodies conjugated to Alexa Fluor^®^ 488 or Alexa Fluor^®^ 594 (Invitrogen, Carlsbad, CA, USA) at a dilution of 1:1000 for 30 min at room temperature. Nuclei were labelled with Hoechst for 10 min. Fluorescence images were captured using the confocal laser scanning microscope (Zeiss LSM 880, Jena, Germany).

### 4.8. Statistical Analysis

Each experiment was performed in triplicate wells and repeated three times. Statistical analyses were performed using GraphPad Prism 6.0 software (GraphPad Software, La Jolla, CA, USA). For multiple comparisons, one-way ANOVA with the Holm-Sidak method was used as indicated. A *p*-value of <0.05 was considered significant. All data were presented as mean ± SEM.

## 5. Conclusions

Dipeptide GP, derived from *Hibiscus sabdariffa* plant cells, inhibited collagen biosynthesis by suppressing Smad2/3 phosphorylation, a major signaling pathway activated by TGF-β1. Additionally, GP abrogated TGF-β1-induced ATF4 expression and the enzymes in the serine/glycine pathway. These findings highlight the anti-fibrotic potential of GP, suggesting its value as a targeted therapeutic intervention for the management and treatment of fibrotic diseases.

## Figures and Tables

**Figure 1 ijms-24-13616-f001:**
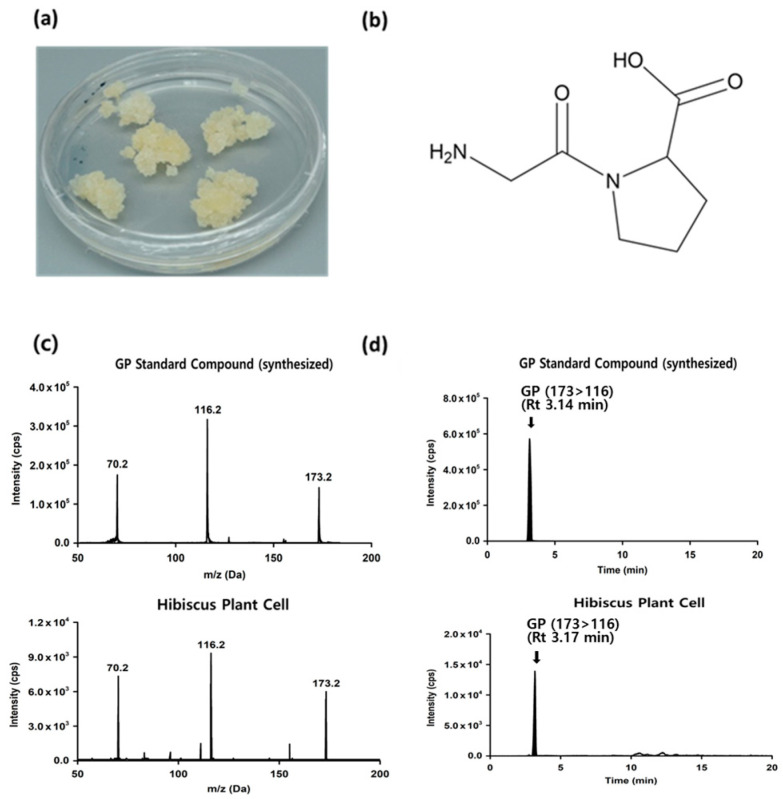
Identification of GP derived from Hibiscus sabdariffa plant cells: (**a**) Plant cells of Hibiscus sabdariffa. (**b**) A structural formula of GP. (**c**) Qualitative analysis of the standard compound GP and the extract of *Hibiscus sabdariffa* plant cells. (**d**) Quantitative analysis of the standard compound GP and the extract of *Hibiscus sabdariffa* plant cells.

**Figure 2 ijms-24-13616-f002:**
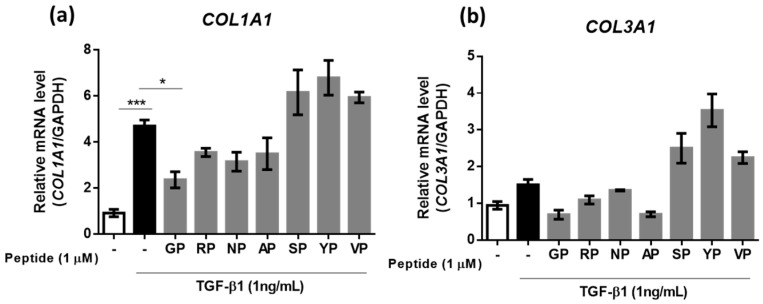
RT-qPCR analysis of the mRNA levels of *COL1A1* and *COL3A1* after treatment with seven dipeptides: GP, RP, NP, AP, SP, YP, and VP. (**a**) *COL1A1*, (**b**) *COL3A1* (n = 3). * *p* < 0.05, *** *p* < 0.001.

**Figure 3 ijms-24-13616-f003:**
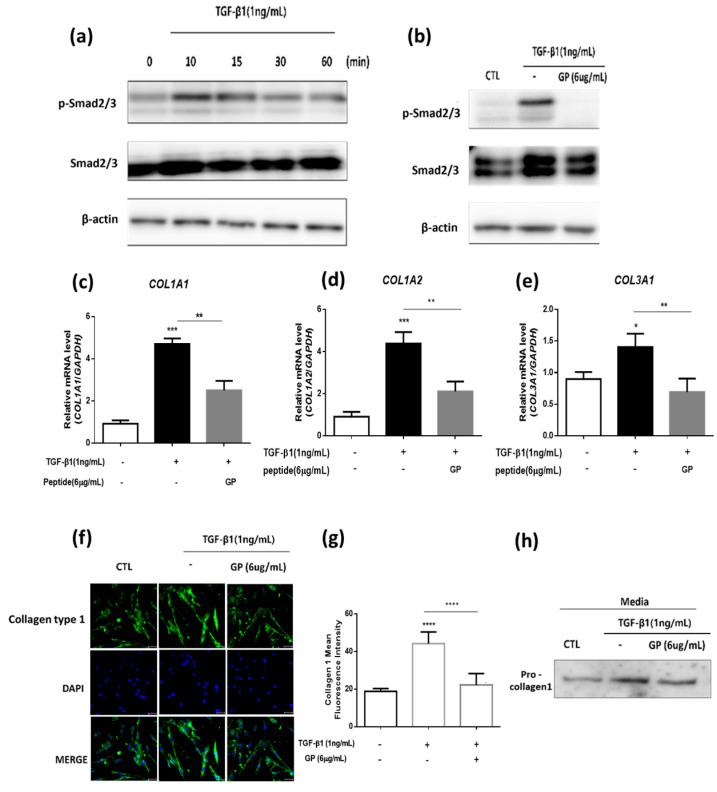
GP suppresses TGF-β1-induced Smad2/3 phosphorylation and collagen synthesis: (**a**,**b**) Western blot analysis of phospho-Smad2/3 and Smad2/3 after treatment with (**a**) TGF-β1 for the indicated times and (**b**) with or without GP. (**c**–**e**) RT-qPCR analysis of mRNA levels of *COL1A1*, *COL1A2*, and *COL3A1* (n = 4). (**f**) Immunocytochemistry staining showing the expression of Collagen type 1 (green). DAPI was used for nuclear staining (blue). Scale bar = 50 µm. (**g**) Quantification of fluorescence intensities (n = 5 images from each group were used for quantification). (**h**) Western blot analysis of pro-collagen1 in the supernatant of HDF treated with TGF-β1 with or without GP. Data are presented as the mean ± SEM. * *p* < 0.05, ** *p* < 0.01, *** *p* < 0.001, **** *p* < 0.0001.

**Figure 4 ijms-24-13616-f004:**
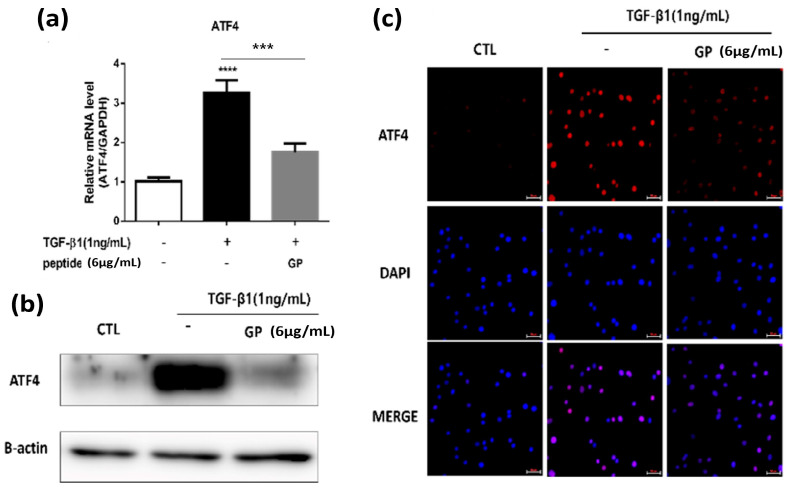
GP downregulates ATF4 expression: (**a**) RT-qPCR analysis of *ATF4* mRNA expression. Data shown as mean ± SEM (n = 4). (**b**) Western blot analysis of ATF4 protein levels. β-actin was used as a loading control. (**c**) Immunofluorescence staining showing the expression of ATF4 (red). DAPI was used for nuclear staining (blue), merged in the overlay image (purple). Scale bar = 50 µm. *** *p* < 0.001, **** *p* < 0.0001.

**Figure 5 ijms-24-13616-f005:**
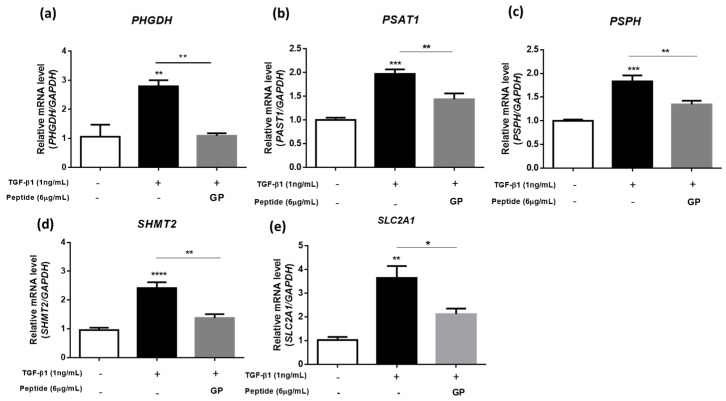
GP decreases the expression of enzymes involved in serine/glycine biosynthesis: (**a**–**e**) RT-qPCR analysis of mRNA levels for *PHGDH*, *PSAT1*, *PSPH*, *SHMT2*, and *SLC2A1* (n = 4). * *p* < 0.05, ** *p* < 0.01, *** *p* < 0.001, **** *p* < 0.0001.

## Data Availability

All related data are within the manuscript.

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
