# Peer review of "The Dipeptide Gly-Pro (GP), Derived from Hibiscus sabdariffa, Exhibits Potent Antifibrotic Effects by Regulating the TGF-β1-ATF4-Serine/Glycine Biosynthesis Pathway"

_ijms, 2023, doi:10.3390/ijms241713616_

Round 1

Reviewer 1 Report

The manuscript discusses the isolation, identification, and mechanistic aspects of antifibrotic dipeptides from Roselle. The design was well supported by the chosen experimental techniques and the results were presented concisely. 

Please do the following minor corrections:

1. Please add explanation/rationale for choosing dipeptides and not larger peptides or any other biomolecules.

2. Please establish the link between peptides and ATF-4.

3. Add references for selecting Proline-based dipeptides for the screening (section 2.1).

4. Introduce HDF full name in section 2.1, paragraph 2.

5. Page numbers are missing for many references, please correct them.

6. In supplementary information figure 1, either remove q-v mass spectra or describe them in the figure legend/description. 

7. In SI figure 1, replace the words "contained" with "present".  

8. Please add retention times to all graphs in SI figure 2.

Author Response

Reviewer 1

Please do the following minor corrections:

  1. Please add explanation/rationale for choosing dipeptides and not larger peptides or any other biomolecules.

A : Thank you for your insightful review. The human skin serves as a physical barrier against exogenous molecules. The size of a molecule significantly determines its ability to penetrate the skin and small molecules such as dipeptides can penetrate the skin barrier. Moreover, given that proline is a primary component of collagen (reference #[17]), and the dipeptides containing proline can influence the biological function of fibroblasts (reference #[20]), our objective was to screen for proline-containing dipeptides with potential antifibrotic effects from the Hibiscus sabdariffa plant cells.

  1. Please establish the link between peptides and ATF4.

A : In our study, we demonstrated that GP effectively counteracted the TGF-β1-induced phosphorylation of Smad2/3 and attenuated ATF4 expression. Given that TGF-β1, a pivotal mediator of tissue fibrosis, can modulate both the transcription and translation of ATF4, and considering Smad3 — a primary downstream regulator of TGF-β1 signaling — interacts with the ATF4 promoter, influencing its transcriptional activity [reference #4, 5], our results indicate that GP can hinder TGF-β1's downstream signaling, leading to the reduction in ATF4 expression. For clarity, we have elaborated on these interactions in the manuscript from lines 190-210.

  1.  Add references for selecting Proline-based dipeptides for the screening (section 2.1).

A : It is well-established that proline and hydroxyproline are amino acids constituting collagen [reference #17-19], and proline-containing dipeptides may influence the biological function of fibroblasts [reference #20]. We have addressed this in the manuscript from lines 68 to 70 and added the relevant references to the reference list.

  1. Introduce HDF full name in section 2.1, paragraph 2.

A : We have added full name for HDF in Section 2.1, Paragraph 2 as suggested.

5. Page numbers are missing for many references, please correct them.

A : We apologize that the reference lists were not complete. We have added the missing page numbers for the references as suggested.

6. In supplementary information figure 1, either remove q-v mass spectra or describe them in the figure legend/description. 

A : We sincerely apologize for the omission in Supplementary Figure 1. We have added the detailed description for each panel in the figure legend.

7. In SI figure 1, replace the words "contained" with "present".  

A : We have replaced the word "contained" with "present" as recommended.

8. Please add retention times to all graphs in SI figure 2.

A : Thank you for your insightful feedback. Based on your suggestion, we have updated Supplementary Figure 2 to include the retention times on all graphs.

Reviewer 2 Report

Review for ijms-2550780

In this research article entitled “The peptide Gly-Pro (GP), derived from Hibiscus sabdariffa, exhibits potent antifibrotic effects by regulating the serine-glycine biosynthesis pathway ”, Kim et al. reported that GP from H. sabdariffa from west africa is a promising therapeutic option for the fibrolitic diseases by regulating ATF4-mediated de novo serine-glycine biosynthesis pathway through the expression of ATF4 itself and attenuation of TGF-b1 induced collagen synthesis.

Hereafter some comments revealed after reviewing the manuscript.

- Use “µg” instead of “ug” in the Figure 1 labelling.

- Promotion of the accumulation of ATF4 by TGF-b1 should be deeply included in the interpretation and extrapolation of the study findings.

- English language is fine just small editing is required. Similarly for the the punctuation.

English language is fine just small editing is required

Author Response

Reviewer 3

1. Use “µg” instead of “ug” in the Figure 1 labelling.

A : We apologize that the unit was not appropriate. We have replaced "ug" with the appropriate “µg” notation.

2. Promotion of the accumulation of ATF4 by TGF-b1 should be deeply included in the interpretation and extrapolation of the study findings.

A : Thank you for your insightful suggestion. As you recommended, we deeply reviewed the relationship among TGF-β1, ATF4, and serine/glycine synthesis pathway and described in the discussion section from lines 192-203.

3. English language is fine just small editing is required. Similarly for the the punctuation.

A : Thank you for your feedback. We have carefully reviewed our manuscript for typos and the punctuation.

Reviewer 3 Report

In this paper, the authors presented the identification of the dipeptide GP as one of the dipeptides derived from Hibiscus sabdariffa. They showed that GP suppresses the phosphorylation of Smad2/3 and decreases the expression of ATF4 (whose upregulation was achieved by TGF-β1) in human dermal fibroblasts. Moreover, they demonstrated that GP decreases the expression of enzymes involved in serine-glycine biosynthesis.

My major concern with this paper is that it is not clear why they selected GP peptide for further studies after the identification of 7 dipeptides in the hibiscus plant cells. Have they tested the effect of the other peptides? Do they have a negative control? These data must be added to the paper.

Other issues:

- In the results chapter the meaning and the importance of each assay must be indicated and discussed.  The results must be better described in terms of significance of contents and importance of the applied  methods. 

- In chapter 2.1 ( Identification of Hibiscus sabdariffa-derived dipeptides), they described the identification of the peptides and their quantification by MS analysis. Also at the end of the chapter they described the mRNA expression levels s of COL1A1 and COL3A1. The two topics must be divided into two chapters, the applied methodologies must be described and the abbreviations must be made explicit. Figure 1 must be divided accordingly.

- Also in chapter 2.3 the data reported in Figure 3 are not well described in the text.

Minor comments:

- the title of the paragraph 2.2 (GP decreased collagen synthesis induced by TGF-β1) must be separated by Figure 1 and must be written in italics

- In supplementary figure 1 the description of some panels is missing

- The MS methods are not adequately described in section 4.1

Please check the spelling of words such as "signifi-cantly" in line 144

Author Response

Reviewer 2

In this paper, the authors presented the identification of the dipeptide GP as one of the dipeptides derived from Hibiscus sabdariffa. They showed that GP suppresses the phosphorylation of Smad2/3 and decreases the expression of ATF4 (whose upregulation was achieved by TGF-β1) in human dermal fibroblasts. Moreover, they demonstrated that GP decreases the expression of enzymes involved in serine-glycine biosynthesis.

My major concern with this paper is that it is not clear why they selected GP peptide for further studies after the identification of 7 dipeptides in the hibiscus plant cells. Have they tested the effect of the other peptides? Do they have a negative control? These data must be added to the paper.

A : We sincerely appreciate your insightful feedback on our paper. In this study, our primary objective was to identify a proline-containing dipeptide with a potential to modulate collagen synthesis in Hibiscus sabdariffa plant cells. Of the seven dipeptides we identified within the Hibiscus sabdariffa plant cell extract, our aim was to pinpoint peptides that could counteract the overexpression of COL1A1 and COL3A1, induced by TGF-β1. Targeting peptides with the potential to regulate collagen synthesis, we exposed human dermal fibroblasts (HDF) to each of the seven dipeptides in the presence of TGF-β1. We then evaluated their capacity to counteract the TGF-β1-driven augmentation in collagen synthesis. Throughout our experiments, fibroblasts that were not exposed to either TGF-β1 or dipeptides served as a negative control. Notably, only GP successfully reduced the expression of COL1A1 and COL3A1, as shown in Figure 2a and 2b. For clarity, we have split the prior Figure 1 into the new Figures 1 and 2, and provided more detailed descriptions in the manuscript from lines 95-102.

Other issues:

- In the results chapter the meaning and the importance of each assay must be indicated and discussed. The results must be better described in terms of significance of contents and importance of the applied methods. 

A : We apologize that our initial description and discussion did not adequately emphasize the significance and implications of our study. We have extensively revised our manuscript, from the abstract to the discussion to ensure a clear interpretation of our data with a comprehensively described methodology.

- In chapter 2.1 (Identification of Hibiscus sabdariffa-derived dipeptides), they described the identification of the peptides and their quantification by MS analysis. Also at the end of the chapter they described the mRNA expression levels s of COL1A1 and COL3A1. The two topics must be divided into two chapters, the applied methodologies must be described and the abbreviations must be made explicit. Figure 1 must be divided accordingly.

A : Thank you for your insightful suggestion. Based on your feedback, we divided the previous chapter 2.1 into new chapters 2.1 and 2.2. Consequently, previous Figure 1 was split into the new Figure 1 and Figure 2.

- Also in chapter 2.3 the data reported in Figure 3 are not well described in the text.

A : We sincerely appreciate your suggestion. We have revised our manuscript to provide a more comprehensive and detailed description of the data. Your insight has significantly contributed to improving the clarity of our manuscript.

Minor comments:

- The title of the paragraph 2.2 (GP decreased collagen synthesis induced by TGF-β1) must be separated by Figure 1 and must be written in italics

A : Since the previous Figure 1 was split into new Figure 1 and Figure 2, the titles for each figure have been revised as follows:

Figure 1. Identification of GP derived from Hibiscus sabdariffa plant cells

Figure 2. RT-qPCR analysis of the mRNA levels of COL1A1 and COL3A1 after treatment with seven dipeptides     

Figure 3. GP suppresses TGF-β1-induced Smad2/3 phosphorylation and collagen synthesis

Furthermore, the titles of each chapter have now been rendered in italics.

- In supplementary figure 1 the description of some panels is missing

A : We sincerely apologize for the omission in Supplementary Figure 1. We have added the detailed description for each panel in the figure legend.

- The MS methods are not adequately described in section 4.1

A : Detailed MS/MS methods with the conditions for instrumental analysis have been added in Supplementary Table 1.

Round 2

Reviewer 2 Report

The manuscript has been revised according to the reviewer comments. Hence, it warrants publication in IJMS.

But I still have one minor comment; it would be better to use "dipeptide" instead of "peptide" for the peptide Gly-Pro (GP) both in the manuscript title and body.

English language is fine, just minor editing is required

Reviewer 3 Report

The authors replied to all my previous comments.